## SCIENCE FORUM

# Imaging methods are vastly underreported in biomedical research

**Abstract** A variety of microscopy techniques are used by researchers in the life and biomedical sciences. As these techniques become more powerful and more complex, it is vital that scientific articles containing images obtained with advanced microscopes include full details about how each image was obtained. To explore the reporting of such details we examined 240 original research articles published in eight journals. We found that the quality of reporting was poor, with some articles containing no information about how images were obtained, and many articles lacking important basic details. Efforts by researchers, funding agencies, journals, equipment manufacturers and staff at shared imaging facilities are required to improve the reporting of experiments that rely on microscopy techniques.

**GUILLERMO MARQUÉS\*, THOMAS PENGO AND MARK A SANDERS**

**\*For correspondence:** marques@umn.edu

**Competing interests:** The authors declare that no competing interests exist.

## Introduction

Over the past three centuries microscopy has evolved from being largely descriptive and qualitative to become a powerful tool that is capable of uncovering new phenomena and exploring molecular mechanisms in a way that is both visual and quantitative (*Trinh and Fraser, 2015*). The optical microscope has also been joined by a wide range of other imaging instruments, and images and data derived from them are crucial to many studies across the life and biomedical sciences.

The authors work at a major imaging facility (https://med.umn.edu/uic) and we are often asked to replicate or expand upon published experiments. However, these experiments are often poorly described, sometimes to the extent that it is not possible to repeat them. Such problems are not limited to microscopy, and concerns about a lack of reproducibility in certain areas of biomedical research have been growing over the past decade (*Ioannidis, 2005*; *Begley and Ellis, 2012*; *Baker, 2016*; *Drucker, 2016*). Causes for concern have included: the substandard characterization of critical resources and reagents, such as

antibodies (*Freedman et al., 2016*; *Schüchner et al., 2020*) and cell lines (*Vaughan et al., 2017*); incomplete reporting of experimental methods and reagents (*Lithgow et al., 2017*); bias (*Macleod et al., 2015*); inadequate statistics (*Benjamin et al., 2018*); and outright fraud (*Bauchner et al., 2018*).

There have been many efforts to address these problems, notably in the area of antibodies and other reagents. As regards incomplete reporting, a number of publishers and funding agencies have signed up to the TOP (Transparency and Openness Promotion) guidelines developed by the Center for Open Science (*Nosek et al., 2015*): signatories to these guidelines commit to promote and enforce good practices of attribution, reporting, data archival, and sharing of research tools (*Sullivan et al., 2019*). To that end, some publishers have established checklists that authors must complete (see, for example, *Development, 2020*; *eLife, 2019*; *Marcus and the whole Cell team, 2016*; *NPG, 2020*), and there is evidence from some areas that these interventions are having a positive effect (*Macleod and the NPQIP*

*Collaboration Group, 2017*; *Han et al., 2017*; *NPG, 2018*). In this article we highlight the need for improved reporting of experiments that involve microscopy.

## Results

To explore the extent and severity of this problem, we examined 240 original research articles in eight journals: Developmental Biology, Development, Developmental Cell, Journal of Cell Biology, Journal of Neuroscience, Nature Immunology, Journal of Immunology, and Biophysical Journal. Just over three-quarters of the papers (185/240 = 77%) had original images, and just over half of the figures in the papers (1439/2780 = 52%) contained images. Most of the images had been acquired by a microscope of some sort, and confocal fluorescence microscopy was the most popular technique: *Supplementary file 1* lists the different imaging techniques used in each of the 185 articles. It should be noted that western blots and similar figures were not considered as images for the purposes of this study (see Materials and methods for details).

Articles about developmental biology and cell biology contained the highest proportion on images, whereas articles about immunology had the lowest (*Table 1*). While the number of figures in an article is a coarse metric that does not address how critical the information provided in a figure is for the conclusions reported in the article, it is an objective and quantifiable metric. It is also important to note that many of the articles contained supplemental videos, further stressing the importance of imaging in biomedical research.

### Methods for imaging experiments are described briefly, if at all

*Table 1* also shows the fraction of the materials and methods section that was devoted to imaging in the 185 articles that contained images. On average just 138 words (7% of the total text in the materials and methods section) was used to describe the details of image acquisition, which seems low given the extent of the imaging results reported, and the fraction for the three developmental biology journals was even lower (5%), despite the high numbers of images in these journals. Moreover, the fractions of text

**Table 1.** Evaluation of the reporting of imaging methods in biomedical journals.
The first column lists journal name, number of articles with images, number of articles evaluated, and the percentage of articles with images. The second column lists the percentage of figures (main and supplemental) that contain original images or quantification of imaging data. The third column lists the percentage of text in the materials and methods sections devoted to imaging (for the 185 articles that contained images). The fourth column lists the percentage of the articles containing images that pass the methods quality test (see Materials and methods for details of this test). Total developmental biology includes three journals (Dev. Biol., Development, and Dev. Cell); total immunology includes two journals (Nature Immunology and J. Immunology). * Five articles containing MRI and X-ray images were not included in the quality evaluation, so the sample for this analysis is 180 papers. *Supplementary file 1* contains a list of all the articles analyzed and details for each article.

| Journal (articles with imaging/total articles, percentage) | Imaging figures (%) | Imaging methods (%) | Pass methods quality (%) |
| --- | --- | --- | --- |
| Developmental Biology (29/30, 99%) | 79 | 4.2 | 3.4 |
| Development (28/28, 100%) | 75 | 7.0 | 14.3 |
| Developmental Cell (32/32, 100%) | 69 | 4.8 | 9.4 |
| J Cell Biology (29/30, 97%) | 72 | 10.1 | 37.9 |
| Nature Immunology (18/29, 62%) | 22 | 5.5 | 11.1 |
| J Immunology (17/31, 55%) | 21 | 2.3 | 5.9 |
| J Neuroscience (18/30, 60%) | 37 | 7.8 | 7.1 |
| Biophysical Journal (14/30, 47%) | 28 | 10.2 | 50.0 |
| Total developmental biology (89/90, 99%) | 74 | 5.2 | 9.0 |
| Total immunolgy (35/60, 58%) | 21 | 4.6 | 8.6 |
| Total (185/240) | 52 | 6.7 | 16.7[*] |

devoted to imaging are over estimates as it was sometimes necessary to include the description of sample preparation in the word count for image acquisition. And somewhat alarmingly, 11 articles (with a total of 56 figures with images) contained no information whatsoever on image acquisition. Setting this group aside, it is possible that an imaging technique can be adequately described in fewer words than, say, a technique in biochemistry, genetics or molecular biology. However, regardless of the word count, it was apparent that many of the articles in our sample did not contain enough information about imaging experiments to allow these experiments to be repeated.

### Few articles contain the information required to replicate the imaging experiments

We also assessed the reporting of three crucial aspects of image acquisition: i) the characteristics of the objective lens used for imaging, a critical determinant of magnification and *optical resolution*; ii) the digitization parameters that determine *image resolution* (image voxel size); iii) the spectral settings for fluorescence imaging that allow efficient signal acquisition and channel discrimination. A combined pass/fail score was then assigned (see 'Imaging materials and methods quality evaluation' in Materials and methods). *Table 1* shows that the overall quality of the information provided is very poor, with less than one in five articles (16.7%) passing the test: the pass rate varied from 3% (for Developmental Biology) to 50% (for the Biophysical Journal).

It must be stressed that our quality test was of very low stringency. The information required to pass was the bare minimum to evaluate and replicate the image, and should not be considered the standard of care. Several proposals over the years have addressed the biological and experimental information that should be collected and reported in the metadata of imaging files (*Huisman et al., 2019*; *Linkert et al., 2010*; *Swedlow et al., 2003*) and more are in development (see, for example, www.doryworkspace. org). These approaches are comprehensive, and extremely valuable for data mining and biological analysis. *Supplementary file 2* is a checklist with the minimal reportable parameters for the two most common types of imaging experiments in our dataset, wide-field fluorescence microscopy and laser-scanning fluorescence confocal microscopy: this checklist is concerned solely with image acquisition parameters, and must be seen as the minimum reporting

guideline for publication. Full imaging metadata reporting requirements that are comprehensive, authoritative, and consensual await development and acceptance by appropriate parties (see 'What to do about it?' below).

It is worth noting that all the examined journals state in their instructions to authors that enough information must be provided to allow critical evaluation and replication of the results. Assessment of the suitability of other segments of the materials and methods section in these publications is beyond the scope of our study. However, spot-checks suggest a much more careful approach to the reporting of molecular biology experiments, with extensive tables of oligonucleotides and antibodies and detailed experimental conditions.

### Reporting of sample preparation methods has improved, but more work is needed

One noticeable improvement brought about by the implementation of the new reporting requirements by some journals is the detailed description of antibodies and their sources. This is a critical aspect of sample preparation and reproducibility in immunofluorescence studies, but by no means the only one. Tissue harvesting and fixation and permeabilization conditions affect sample integrity (*Schnell et al., 2012*). Probing and washing steps and the nature of the mounting/imaging medium critically influence the quality of the images obtained (*Boothe et al., 2017*; *Fouquet et al., 2015*). Quantifying the extent to which these parameters are reported goes beyond the scope of the present study, but we did look at sample preparation for electron microscopy (EM) images because minor differences in sample processing can result in major differences in tissue ultrastructure that are harder to notice by optical methods. We found that only 4 of the 14 papers with EM images included sufficient detail to allow sample preparation to be replicated. The issues ranged from not giving any details, to pointing to inappropriate references or lacking important method details such as durations, concentrations, pH and so on. The complete reporting of sample preparation methods in optical microscopy is equally critical, particularly as optical 'super resolution' techniques begin to bridge the gap between optical and electron microscopy (*Gwosch et al., 2020*; *Pereira et al., 2019*).

While reporting of sample preparation details has improved, the adoption of STAR methods by Developmental Cell in 2017 has not resulted

in adequate reporting of image acquisition details. In our dataset only 9% of the Dev. Cell articles provided enough experimental information to attempt replication of the imaging experiment. Similarly, despite Nature Immunology using the Nature Research Life Sciences Reporting Summary, only 11% of the articles passed the test (*Table 1*). It appears these new reporting requirements do not cover imaging appropriately.

### Image processing and analysis are rarely described in detail

A final observation is that many articles contained little or no information about the procedures used for image processing, analysis, or quantification. We have not performed a quantitative analysis of this area because of the difficulty in creating a scoring matrix for a widely variable set of analysis procedures. The critical issue is that identical images can be processed in multiple ways, and many different algorithms can be used for segmentation, so the resulting quantification can be different (*Botvinik-Nezer et al., 2020*). Without knowing the processing steps the image went through between acquisition and quantification, it is not possible to replicate the quantitative analysis. Proper reporting of image analysis requires a detailed description of the ways the image has been processed and the parameters used, followed by details of segmentation and quantification. It is imperative that researchers keep track of the steps and report them fully and accurately in their publications: see *Hofbauer et al., 2018* for a good example of how to report these details. Unfortunately, most image-analysis programs do not record these processing steps in the metadata of images. An exception to this is OMERO (*Allan et al., 2012*), a free, open-access image repository that allows image processing and analysis while keeping track of the image manipulations.

## Discussion

Our study raises several issues. First, when it comes to imaging the "reproducibility crisis" is really a "preproducibility" crisis: in general there is not enough information in an article for anyone who wants to repeat an imaging experiment (*Stark, 2018*). This is a serious problem that causes unnecessary waste of researchers' time and resources trying to figure out how an experiment was actually done, before even attempting to replicate it. Also, given the role of

unexpected variability on experimental results, exacting descriptions of the materials used and procedures followed are essential to ensure reproducibility (*Lithgow et al., 2017*; *Niepel et al., 2019*).

Second, it is puzzling that authors devote a substantial effort to document other experimental techniques, but fail to do so for the basics of imaging. We do not have a good explanation for this, but it is worth noting that while formal training in biochemistry, genetics, and molecular and cellular biology is mandatory in most undergraduate and graduate biomedical programs, microscopy and imaging are rarely part of the curriculum. Our suspicion is that authors are not quite sure as to what really matters in an imaging experiment (*Jost and Waters, 2019*; *North, 2006*). It is interesting to note that the Nature Research Life Sciences Reporting Summary includes specific and detailed questionnaires for antibodies, cell lines, statistical analysis, ChIP-Seq, flow cytometry, and MRI, but not for optical imaging (https://www.nature.com/documents/nr-reporting-summary-flat.pdf). Similarly, in its editorial policies Nature encourages, but does not require, reporting of critical image acquisition and processing parameters (https://www.nature.com/nature-research/editorial-policies/image-integrity#microscopy). This does not seem to be enough to ensure accurate reporting of imaging procedures.

Third, it is hard to understand how reviewers and editors can accurately evaluate the results of a manuscript when there is not enough information on how the experiments were performed. It seems reviewers take the reported results at face value, without much consideration to the limitations that the experimental procedures may impose on the data.

Fourth, it is apparent that editors and publishers are not enforcing the requirements they have mandated. As an example, for microscopy experiments the Journal of Cell Biology requires that the numerical aperture (NA) of the objective lens used be reported (*Rockefeller Press, 2019*): however, almost a quarter (7/29) of the articles from this journal in our dataset failed to disclose the NA of the objective lens used.

Fifth, while we have not completed an exhaustive analysis of all biomedical areas, we are confident this problem extends to other disciplines, such as physiology and cancer biology.

## What to do about it?

We believe that the massive underreporting of the details of microscopy experiments needs to be addressed urgently. As Lithgow et al. wrote: "We have learnt that to understand how life works, describing how the research was done is as important as describing what was observed" (*Lithgow et al., 2017*).

Authors need to improve their understanding of the imaging techniques they use in their research, and reviewers and editors need to insist that enough information is given to evaluate and replicate experimental imaging data. Mandatory deposit of original image files (including accurate metadata; *Linkert et al., 2010*) in a repository would be a step in the right direction. This approach was the basis for the JCB Dataviewer that was tested and ultimately discontinued by the Journal of Cell Biology (*Hill, 2008*). Existing image repositories such as OMERO for microscopy images (https://www.openmicroscopy.org/omero/) and the more generic BioImage Archive (https://www.ebi.ac.uk/bioimage-archive/) could serve this purpose (*Allan et al., 2012*; *Ellenberg et al., 2018*). More specialized resources – such as the Brain Research Microscopy Workspace (www.doryworkspace.org) or the Cell Image Library (www.cellimagelibrary.org) – could also contribute to the development of minimum reporting guidelines for images. These guidelines should cover the technical details of image acquisition and the biological information required to provide adequate context (*Huisman et al., 2019*). Vendors of imaging instrumentation, in particular microscopes, must also strengthen and standardize the procedures by which metadata is recorded in acquired images to facilitate its retrieval.

While editors, researchers and vendors have responsibilities in this area, scientists in shared imaging facilities (like ourselves) have a central role to play in ensuring accurate reporting of critical imaging parameters. This role includes educating clients on the relevant variables that affect their experimental results and on the importance of faithfully reporting that information. On a more immediate and practical matter, staff at such facilities can provide off-the-shelf descriptions and vet the methods section of manuscripts. For example, we have developed MethodsJ, a FIJI script that extracts metadata (microscope model, objective lens magnification, NA, excitation and emission wavelengths, exposure time) from a light microscopy image using the Bio-Formats library and generates text can be used in the materials and methods section of an article (*Schindelin et al., 2012*; *Linkert et al., 2010*). While MethodsJ can reliably retrieve some of the critical parameters (objective lens magnification, NA, voxel size), it is more difficult to retrieve the excitation and emission wavelengths for fluorescence: the adoption of standards for metadata by different manufacturers would also make MethodsJ more robust. The code for MethodsJ is available at https://github.com/tp81/MethodsJ (copy archived at https://github.com/elifesciences-publications/MethodsJ; *Pengo, 2020*) and some example output is shown in *Supplementary file 3*.

On a broader scale, professional societies such as the Microscopy Society of America (MSA), Bio Imaging North America (BINA), and the Association of Biomolecular Resource Facilities (ABRF) could develop an appropriate checklist for the reporting of imaging experiments in coordination with publishers. These organizations will also be instrumental in the dissemination of these standards through meetings, workshops and educational activities. Depositing any standards in the FAIRsharing repository (fairsharing.org; *Sansone et al., 2019*) could also help with adoption by the community.

Ultimately, the enforcement of standards will have to fall on publishers and funding agencies. In the US the National Institutes of Health and the National Science Foundation already require that experimental methodology is reported and made publicly available, and many journals have similar requirements. It seems that further educational efforts are needed to ensure researchers report their methods fully, and a more proactive approach by journal editors seems to have a beneficial effect on the rigor of experimental reporting (*Miyakawa, 2020*). Endorsement and support of the image repositories by funders and publishers and mandatory deposit of fully annotated published images will be instrumental in ensuring proper reporting of imaging data and improving the reproducibility of biomedical research.

## Materials and methods

### Article selection

Issues from late 2018 and early 2019 of Development, Developmental Biology, Developmental Cell (developmental biology journals), Journal of Immunology, Nature Immunology (immunology journals), Journal of Cell Biology, Biophysical

Journal, and Journal of Neuroscience were randomly selected. All original research articles in an issue were selected up to 30 articles per publication. If there were less than ~30 articles per issue, consecutive issues were used until reaching that number. Reviews, commentaries and editorials were not used in this analysis. We analyzed a total of 240 articles.

### Figure classification

All content of the articles was included in the analysis, including supplementary text and figures. For the purposes of this study, we considered all optical and electron microscopy images and all preclinical animal imaging when assessing the extent of imaging used. Images of western blots, polyacrylamide gels, phosphorimagers and the like were not included. The number of figures in the main text and supplemental information were assigned as imaging figures if any of the panels in the figure contained an original image (not a cartoon or line diagram, schematics etc.). Figures that contained quantification of imaging data were also considered imaging figures, even if no images were displayed. Conversely, plots, graphs, and figures rendering data derived form other primary techniques, such as modeling, flow cytometry or western blots, were not considered imaging figures. To control for over estimation of the imaging content of an article, the main figures in 12 articles of an issue of Development were quantified on a panel-by-panel basis. Each panel in a figure (A, B, etc.) was assigned to imaging or not as above. The 12 articles thus analyzed contained 94 primary figures and 641 panels. 85 of the figures (90%) and 558 of the panels (87%) were classified as imaging. As we did not find a pronounced difference in the extent of imaging usage between the two methods of evaluation (whole figure and panel classifications), we stuck to the simpler whole figure classification for our analysis. All articles in a journal were used to determine the ratio of imaging to total figures, including those that did not contain any imaging data.

### Imaging materials and methods quantification

The whole materials and methods section of the main text and supplementary materials were considered. The STAR Methods section (excluding the key resources table) was included for Developmental Cell. The Life Sciences Reporting Summary in Nature Immunology was not

included because it contains a lot of boilerplate text that is not relevant in all articles. All the text was put together and subject to word counting in Word (Microsoft). All the portions of the materials and methods section that contained information on image acquisition were extracted and separately counted. Text devoted to sample preparation prior to image acquisition (for example, antibodies used, or immunofluorescence protocols) or to image analysis was not counted unless inextricably interwoven with the image acquisition description. Similarly, text about western blots etc. was not counted (as western blots etc. were not considered as images for the purposes of our study). To quantify the ratio of text about imaging to total materials and methods text, only articles that contained imaging were considered (185 out of the initial 240 articles).

### Imaging materials and methods quality evaluation

Knowing the imaging word count in Materials and methods turned out not to be very informative, as some texts were devoid of usable information (see below). To address this issue, the imaging information was evaluated qualitatively in three separate aspects.

First, for proper evaluation and replication of any optical microscopy experiment the resolution and magnification used are essential. In addition, the degree of chromatic and planar aberration correction is needed for multichannel fluorescence imaging and quantitative microscopy (*Ross et al., 2014*). This requires reporting objective lens correction, magnification, and numerical aperture (NA). In order to pass this section both NA and magnification had to be reported.

Second, the parameters for digitization were evaluated. Planar optical resolution provided by the objective lens can only be adequately captured in the image if the digitization sampling pitch is correct (*Stelzer, 1998*). When a three-dimensional image is acquired, the interval between planes (Z step) is also needed in combination with the NA of the objective lens and the size of the confocal aperture (if used) to determine if axial digital sampling is adequate. Planar sampling parameters can be best reported as the pixel size of the digital file, but can also be derived from the total magnification and camera model in wide field images, or a combination of frame size and laser zoom for confocal microscopy. Reporting any of these parameters is enough to pass the digitization section. When

three-dimensional images are reported, the voxel size or the Z step was also required to pass this section.

Third, fluorescence microscopy is by far the most common imaging technique in our dataset (confocal fluorescence microscopy is used in 62% of the imaging articles, and wide-field fluorescence microscopy is used in 40%). Adequate replication and evaluation of this technique requires knowledge of the excitation and emission bands used to capture fluorescence (*Waters and Wittmann, 2014*; *Webb and Brown, 2013*). This is particularly critical in multi-channel acquisitions. When fluorescence imaging experiments were reported, the wavelengths used for excitation and emission with the different fluorochromes were needed to pass this section. Passing the objective lens section plus one of the other two sections is necessary for a global passing grade in this qualitative assessment.

When electron microscopy was reported, electron microscopy imaging was evaluated on a case-by-case basis by one of the authors (MAS), who has more than 30 years experience in the technology. While the instrument manufacturer and model may be relevant, the acquisition settings, including accelerating voltage, gun bias, magnification, and spot size must also be conveyed. These parameters determine contrast, resolution, and signal to noise ratio (*Egerton, 2014*). Additionally, sample preparation including fixation, dehydration, embedding and sectioning can demonstrably impact the outcome of the study and should also be reported. Evidence of adequate fixation includes uniform presence of ground structure in all organellar compartments, membrane bilayers intact and parallel, mitochondria and endoplasmic reticulum not distended nor extracted with membrane structures intact (*Dykstra and Reuss, 2003*).

Whole animal luminescence and fluorescence imaging methods had to provide the digitization information, and if appropriate the spectral bands used as described above for microscopy to obtain a passing grade. *Supplementary file 1* lists the score for each of these categories in columns N (objective lens), O (digitization), P (spectral settings), and Q (EM); together with the final score (R, Global).

Five articles that contained only Magnetic Resonance Imaging or X-ray imaging were excluded from this analysis. While these imaging modalities should be subject to the same good reporting practices, they fall away from our area of expertise. This leaves 180 imaging articles subject to evaluation of the quality of image acquisition reporting (listed as Eligible articles in column O of the Summary Tab in *Supplementary file 1*).

To illustrate the quality evaluation method, the following three publications are examples of failing, passing but incomplete, and good reporting. The average number of words about imaging in the materials and methods sections for articles with images was 138 (*Supplementary file 1*). The first example ([doi.org/10.1016/j.devcel.2018.08.006](doi.org/10.1016/j.devcel.2018.08.006)) contains 121 words about imaging methods, but these words fail to provide the basics of image acquisition: "Samples were imaged with a Nikon Eclipse inverted microscope with A1R confocal running NIS Elements and images were analyzed with Fiji (*Schindelin et al., 2012*). Super-resolution microscopy was performed with a Nikon A1 LUN-A inverted confocal microscopy. C2C12 myoblasts were differentiated for two days and immunostained with custom Myomaker antibodies (Gamage et al., 2017) and Myomerger antibodes described above. Images were acquired with a 100X objective NA1.45 at four times the Nyquist limit (0.03 µm pixel size). Z- stacks were acquired using a 0.4 AU pinhole yielding a 0.35 µm optical section at 0.1 µm intervals and 2X integration to avoid pixel saturation. Images were deconvolved with NIS elements using 15 iterations of the Landweber method. Images shown are a single focal plane." No information is provided on objective lens, digitization, or fluorescence parameters for confocal microscopy. The super-resolution acquisition experiment is correctly described in objective lens and digitization, but fails in spectral parameters details. Out of 12 imaging figures in this article, just one panel is super-resolution, no information on how the other images were acquired.

In the second example ([doi.org/10.1016/j.devcel.2018.07.008](doi.org/10.1016/j.devcel.2018.07.008)) 79 words are enough for a passing grade: "Quantitative Microscopy Confocal images were acquired using a Nikon Eclipse Ti-E microscope (Nikon Corp.) equipped with a swept-field confocal scanner (Prairie Technologies), a 100x Plan Apochromat objective (NA 1.45) and an Andor iXon EM-CCD camera (Andor). Widefield images were acquired with a Nikon Eclipse Ti-E microscope (Nikon Corp.) equipped with a 100x Plan Apochromat objective (NA 1.40) and an Andor Xyla 4.2 scientific CMOS camera (Andor). Laser intensity and exposures were identical for all images that were quantitatively compared." The

fluorescence channel information is missing, but objective lens and digitization are properly reported.

Lastly, this example (doi.org/10.1016/j.bpj.2018.12.012) passes all three aspects with 136 words: "Cell observation. Fluorescence images were obtained by using an inverted microscope (IX81-ZDC2; Olympus, Tokyo, Japan) equipped with a motorized piezo stage and a spinning disc confocal unit (CSU-X1-A1; Yokogawa, Musashino, Japan) through a 60 [sic] oil immersion objective lens (numerical aperture 1.35; UPLSAPO 60XO; Olympus, Tokyo, Japan). PtdInsP3-GFP was excited by a 488 nm laser diode (50 mW). The images were passed through an emission filter (YOKO 520/35; Yokogawa) and captured simultaneously by a water-cooled electron-multiplying charge-coupled device camera (Evolve; Photometrics, Huntington Beach, CA). Time-lapse movies were acquired at 10 s intervals at a spatial resolution of dx = dy = 0.2666 μm and dz = 0.5 μm using z-streaming (MetaMorph 7.7.5; MetaMorph, Nashville, TN). Cells observed in the microfluidic chamber are acquired at a spatial resolution of dx = dy = 0.2222 μm and dz = 0.2 μm".

## Acknowledgements
Dr. Daniel Ortuño-Sahagún was a key catalyst for this project. Shihab Ahmed and Neah Roakk helped greatly with the gathering of the journal articles.

**Guillermo Marqués** is at the University Imaging Centers (UIC) and the Department of Neuroscience, University of Minnesota, Minneapolis, United States
marques@umn.edu
https://orcid.org/0000-0003-1478-1955

**Thomas Pengo** is at the University of Minnesota Informatics Institute (UMII), University of Minnesota, Minneapolis, United States
https://orcid.org/0000-0002-9632-918X

**Mark A Sanders** is at the University Imaging Centers (UIC) and the Department of Neuroscience, University of Minnesota, Minneapolis, United States
https://orcid.org/0000-0001-7550-5255

*Author contributions:* Guillermo Marqués, Conceptualization, Data curation, Formal analysis, Methodology, Writing - original draft, Writing - review and editing; Thomas Pengo, Conceptualization, Software, Writing - review and editing; Mark A Sanders, Conceptualization, Formal analysis, Writing - review and editing

*Competing interests:* The authors declare that no competing interests exist.

## Funding
No external funding was received for this work.

## Decision letter and Author response
Decision letter https://doi.org/10.7554/eLife.55133.sa1
Author response https://doi.org/10.7554/eLife.55133.sa2

## Additional files
### Supplementary files
• Supplementary file 1. A list of all the articles analyzed for *Table 1* and details for each article.

• Supplementary file 2. Reporting guidelines for wide-field fluorescence microscopy and laser-scanning fluorescence confocal microscopy.

• Supplementary file 3. Example output from MethodsJ.

• Transparent reporting form

## Data availability
All data generated or analyzed during this study are included in the manuscript and supporting files.

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
