## [Decision Letter]

Thank you for submitting your article "Imaging in Biomedical Research: An Essential Tool with No Instructions" for consideration by *eLife*. Your article has been reviewed by three peer reviewers, and the evaluation has been overseen by the *eLife* Features Editor (Peter Rodgers). The following individuals involved in review of your submission have agreed to reveal their identity: Claire Brown (Reviewer #1); Elisabeth M Bik (Reviewer #3).

Summary:

This paper brings attention to the ongoing problem of lack of detail in the Materials and methods section of paper to reproduce imaging experiments. The authors do a nice study of some select literature to highlight the breadth and depth of the problem. This is an important issue that needs to be discussed in the community and dealt with and this article will help push towards correcting the problem in the field. However, a number of points need to be addressed to make the article suitable for publication.

Essential revisions:

1) The Appendix should include a clear definition of what counted as an "image" for this paper. Where Western blots or other non-microscopy photos included in this definition? Did the authors count line graphs, flow cytometry panels, or spectra as "image" or not?

Here are three examples to illustrate how the short definition of an image for this paper does not appear to match the data provided in the Source data file:

# DOI:10.1016/j.bpj.2018.12.008 is listed to include 2 images in 7 figures. But I would not be able to know which 2 were counted as images. Figure 2 and 3 contain microscopy photos, but Figures 4 and 5 also appear to contain some imaging data, albeit not a real photo but more a spectrum? Which ones were counted as an image?

# DOI:10.1038/s41590-018-0203-2 has 5 figures with images. Some panels of those figures were microscopy images, but others were immunoblots. Were both types of images evaluated for this paper or only the microscopy photos?

# DOI:10.1083/jcb.201710058 has 8 figures with photographic images, not 7 as listed in the Source Data File. That might suggest that the authors did not include Figure 4 (which has Western blot photos but not microscopy photos) in their analysis, and that they mainly focused on microscopy photos.

2) Because it is not clear what the authors counted as an "image", it is also not clear what was counted as the amount of text in the methods devoted to images. For example, if a paper contained both immunoblots as well as microscopy photos, did the authors only look for the microscopy paragraphs in the Methods, or were the immunoblot (Western blot) paragraphs also counted? Often, Western blot imaging is not described in the Methods because it is not very important how a photo is made, and a smartphone camera might suffice. For microscopy, the settings and details are much more important than for Western blots. It would be helpful if the Introduction / Discussion would better clarify that this paper focused on settings for microscopy photos, not on those for other types of photos. The examples in the Appendix were very helpful to clarify, but already defining this at the beginning of the paper would be better to avoid confusion.

3) The paper lists that 185 articles containing images were evaluated, but in the legend of Table 1, suddenly 4 papers with MRI images are removed. This is not mentioned in the main text. It would be better to describe (e.g. in the Appendix) that MRI images were not included in the definition of 'images' and list 181 as the total of papers analyzed or to include them in all analyses.

4) There is mention of the importance of sample preparation in the examples provided about EM microscopy and how that was evaluated but little is mentioned about optical microscopy and sample preparation. This is very important for quantitative and reproducible fluorescence imaging as well. Details that are important include catalog numbers and lot numbers for reagents and validating staining methods or functionality of tagged proteins.

The manuscript would be strengthened considerably if the authors could extend their analysis of the 240 papers in their sample to determine how rigorous and detailed they are when it comes to sample preparation. If this is not possible, the authors should mention the importance of reporting the details of sample preparation at appropriate places in their manuscript, and acknowledge that the percentages for imaging methods that they quote do not include sample preparation.

5) Similarly, image analysis is mentioned but not covered in detail. I think it is critical to document image analysis steps. Without this imaging experiments cannot be reproducible. Details that are important include data providence, the importance of retaining raw image data, the importance of documenting each analysis step including software versions and operating system. The OMERO figure software could be mentioned here as it offers data analysis tracking including for multiple people manipulating the same data set.

As in the previous point, the manuscript would be strengthened considerably if the authors could extend their analysis to include image analysis. If this is not possible, the authors should mention the importance of reporting the details of image analysis at appropriate places in their manuscript, and acknowledge that the percentages for imaging methods that they quote do not include image analysis.

6) The authors identify that incomplete reporting of imaging methods is a problem that a number of members of the community must address, including imaging facilities and journals. Having checklists is a great start. A preliminary checklist for critical image acquisition parameters should be included as a supplemental material for this publication. Perhaps the list of metadata pulled from images with the methodsJ plugin has that information already? Also, the parameters for image analysis (described above) should be included in the checklist. What else might journals do besides checklists (maybe use AI to screen the Materials and methods section for minimal criteria during the submission process?) And what more could core facilities of national and international communities (such as BINA and ABRF) do (maybe develop and disseminate training programs for researchers and reviewers?)

7) Another key stakeholder that is only briefly mentioned is the microscope manufacturers. They need to be guided and encouraged to include significant metadata as part of acquired images and make that information accessible to the end user on multiple different software platforms. A comment on this is needed in the discussion.

8) The authors mention funders as enforcers. Please expand on this - what could funders do to improve the reporting of imaging methods in papers?

9) A large part of the challenge in this field is to raise awareness. This article does just that. So it is also important to point people to a range of emerging resources in this area. Suggestions include:

# OMERO should be mentioned as a potential resource for centralized data management, accessibility and sharing.

# I would also suggest you reference work being done in this area by the 4D Nucleome project: "Minimum Information guidelines for fluorescence microscopy: increasing the value, quality, and fidelity of image data" The 4DN-OME model extension and metadata guidelines: https://arxiv.org/abs/1910.11370

# The brain microscopy initiative is asking for feedback on image metadata standards they are developing. https://brainmicroscopyworkspace.org/feedback.

# BioImage Archive is meant to be a repository for published microscopy image data sets so they can be used by the broader microscopy community. https://www.ebi.ac.uk/about/news/press-releases/bioimage-archive-launch

---

## [Author Response]

Essential revisions1) The Appendix should include a clear definition of what counted as an "image" for this paper. Where Western blots or other non-microscopy photos included in this definition? Did the authors count line graphs, flow cytometry panels, or spectra as "image" or not?Here are three examples to illustrate how the short definition of an image for this paper does not appear to match the data provided in the Source data file:# DOI:10.1016/j.bpj.2018.12.008 is listed to include 2 images in 7 figures. But I would not be able to know which 2 were counted as images. Figure 2 and 3 contain microscopy photos, but Figures 4 and 5 also appear to contain some imaging data, albeit not a real photo but more a spectrum? Which ones were counted as an image?# DOI:10.1038/s41590-018-0203-2 has 5 figures with images. Some panels of those figures were microscopy images, but others were immunoblots. Were both types of images evaluated for this paper or only the microscopy photos?# DOI:10.1083/jcb.201710058 has 8 figures with photographic images, not 7 as listed in the Source Data File. That might suggest that the authors did not include Figure 4 (which has Western blot photos but not microscopy photos) in their analysis, and that they mainly focused on microscopy photos.

Image definition and western blots.

We did not consider western and other so-called molecular imaging techniques (SDS-PAGE, Northern…) in our analysis. This was rather embarrassingly not stated in our manuscript, and it was clearly a source of confusion. We have now added a line in the main text and a paragraph in the Appendix making this exclusion criterion explicit.

Line graphs or spectra are not images and are then not considered, we have made this clearer in the Appendix.

The specific examples of figures that were not counted as imaging in DOI:10.1016/j.bpj.2018.12.008 (Deng, 2019, Figures 4 and 5) and are referred to as “spectrum” by a reviewer are, to the best of my understanding, the results of a computer simulation displayed with a colorful look up table, and do not represent actual imaging data. I have reviewed the images again and I still think they are not real imaging data.

2) Because it is not clear what the authors counted as an "image", it is also not clear what was counted as the amount of text in the methods devoted to images. For example, if a paper contained both immunoblots as well as microscopy photos, did the authors only look for the microscopy paragraphs in the Methods, or were the immunoblot (Western blot) paragraphs also counted? Often, Western blot imaging is not described in the Methods because it is not very important how a photo is made, and a smartphone camera might suffice. For microscopy, the settings and details are much more important than for Western blots. It would be helpful if the Introduction / Discussion would better clarify that this paper focused on settings for microscopy photos, not on those for other types of photos. The examples in the Appendix were very helpful to clarify, but already defining this at the beginning of the paper would be better to avoid confusion.

Imaging text

We have clarified that methods descriptions for western etc., sample prep, and image analysis are not considered in the amount of text classified as Imaging. There are exceptions due to the ways the methods are written, where one cannot cleanly extricate the imaging methods from sample prep or analysis, but these instances are a small portion of the total. At any rate, this would only cause an over count of the imaging materials and methods. We have clarified this point in the main text and the Appendix.

The focus of this article is not only microscopy, although the majority of the images are microscopy. We prefer to keep the broader imaging scope, because the reporting criteria for other biomedical imaging are similar: resolution and magnification, digitization parameters and spectral settings have to be properly reported in animal imagers the same as in microscopes.

3) The paper lists that 185 articles containing images were evaluated, but in the legend of Table 1, suddenly 4 papers with MRI images are removed. This is not mentioned in the main text. It would be better to describe (e.g. in the Appendix) that MRI images were not included in the definition of 'images' and list 181 as the total of papers analyzed or to include them in all analyses.

Articles excluded from quality evaluation

We think that the articles that contain only MRI (4, all in J. Neurosci.) and X-Ray imaging (1, in Nature Immunology) should be counted as imaging articles, contributing to the 185 imaging articles of the total 240 articles evaluated. Just because we are not comfortable evaluating them for accuracy of reporting does not mean they don’t contain images. However they are not counted in the denominator of the quality evaluation rate, articles that pass the quality criteria/imaging articles. This denominator is 180 eligible imaging articles that we evaluated for quality of reporting. These are listed under **Eligible articles** in column O of the Summary tab of the Source Data File. We have made this clearer in the Appendix and added a footnote to the table legend to this effect.

4) There is mention of the importance of sample preparation in the examples provided about EM microscopy and how that was evaluated but little is mentioned about optical microscopy and sample preparation. This is very important for quantitative and reproducible fluorescence imaging as well. Details that are important include catalog numbers and lot numbers for reagents and validating staining methods or functionality of tagged proteins.The manuscript would be strengthened considerably if the authors could extend their analysis of the 240 papers in their sample to determine how rigorous and detailed they are when it comes to sample preparation. If this is not possible, the authors should mention the importance of reporting the details of sample preparation at appropriate places in their manuscript, and acknowledge that the percentages for imaging methods that they quote do not include sample preparation.

Sample preparation quality evaluation

We could not agree more strongly with the reviewer that sample preparation is a critical aspect of image acquisition. And that accurate description of what was done is critical for reproducibility. However this analysis falls way beyond the scope of this article, and would require a very substantial effort. We have added content emphasizing the importance of this topic, with some critical references.

5) Similarly, image analysis is mentioned but not covered in detail. I think it is critical to document image analysis steps. Without this imaging experiments cannot be reproducible. Details that are important include data providence, the importance of retaining raw image data, the importance of documenting each analysis step including software versions and operating system. The OMERO figure software could be mentioned here as it offers data analysis tracking including for multiple people manipulating the same data set.As in the previous point, the manuscript would be strengthened considerably if the authors could extend their analysis to include image analysis. If this is not possible, the authors should mention the importance of reporting the details of image analysis at appropriate places in their manuscript, and acknowledge that the percentages for imaging methods that they quote do not include image analysis.

Image processing and analysis reporting evaluation

Again, a topic dear and near to our hearts, but a completely different paper. We have expanded a little bit on the topic and pointed to one of the articles in our dataset that does a really good job of describing the image processing and analysis workflow. We have also incorporated a recent reference that highlights the perils of poor analysis reporting, as well as the requested OMERO reference.

6) The authors identify that incomplete reporting of imaging methods is a problem that a number of members of the community must address, including imaging facilities and journals. Having checklists is a great start. A preliminary checklist for critical image acquisition parameters should be included as a supplemental material for this publication. Perhaps the list of metadata pulled from images with the methodsJ plugin has that information already? Also, the parameters for image analysis (described above) should be included in the checklist. What else might journals do besides checklists (maybe use AI to screen the Materials and methods section for minimal criteria during the submission process?) And what more could core facilities of national and international communities (such as BINA and ABRF) do (maybe develop and disseminate training programs for researchers and reviewers?)

A. Imaging checklist and MethodsJ reporting

The variety of imaging methodologies we encountered in our study makes not feasible to provide a specific checklist for each one of them, and as we state in our manuscript, this prescriptive task should be left to the appropriate organizations that can provide a broader, authoritative consensus. At the same time we realize that it can sound glib to state something is wrong without providing a template on how to do it right. We have provided a narrow checklist for the two most common imaging techniques in out dataset, fluorescence laser scanning confocal microscopy and fluorescence wide field microscopy (Appendix 2). We are also providing as Appendix 3 the MethodsJ output of multiple imaging files from different methods and vendors. This appendix demonstrates both the capabilities of the script and the need for a more consistent reporting of metadata by vendors (see 7 below). We have added the appropriate language in the text regarding both Appendices.

B. What else can be done by journals and professional societies?

These are critical parts of the solution and we have expanded a little bit on them. Education and dissemination will be the essential role of professional societies. The journals need to be more proactive in their requests for primary experimental data.

7) Another key stakeholder that is only briefly mentioned is the microscope manufacturers. They need to be guided and encouraged to include significant metadata as part of acquired images and make that information accessible to the end user on multiple different software platforms. A comment on this is needed in the discussion.

Microscope manufacturers

Point well taken. We have added some text on their role to ensure accurate, uniform recording of essential metadata in the image files. This partially feeds from the MethodsJ test runs in Appendix 3, that show the gold standard for imaging metadata extraction, Bio-Formats, is sometimes incapable of interpreting the manufacturer’s files.

8) The authors mention funders as enforcers. Please expand on this - what could funders do to improve the reporting of imaging methods in papers?

Funding

We have expanded a little bit on this, but this is necessarily speculative. Although if granting agencies put some teeth on their reporting mandates things would change rapidly.

9) A large part of the challenge in this field is to raise awareness. This article does just that. So it is also important to point people to a range of emerging resources in this area. Suggestions include:# OMERO should be mentioned as a potential resource for centralized data management, accessibility and sharing.# I would also suggest you reference work being done in this area by the 4D Nucleome project: "Minimum Information guidelines for fluorescence microscopy: increasing the value, quality, and fidelity of image data" The 4DN-OME model extension and metadata guidelines: https://arxiv.org/abs/1910.11370# The brain microscopy initiative is asking for feedback on image metadata standards they are developing. https://brainmicroscopyworkspace.org/feedback.# BioImage Archive is meant to be a repository for published microscopy image data sets so they can be used by the broader microscopy community. https://www.ebi.ac.uk/about/news/press-releases/bioimage-archive-launch

Raising awareness

We have incorporated those excellent suggestions.